# Effects of Different Types of Soil Management on Organic Carbon and Nitrogen Contents and the Stability Index of a Durum Wheat–Faba Bean Rotation under a Mediterranean Climate

Luigi Tedone *[ID], Leonardo Verdini and Giuseppe De Mastro

Department of Soil, Plant and Food Sciences, University of Bari "Aldo Moro", Via G. Amendola 165/a, 70126 Bari, Italy; giuseppe.demastro@uniba.it (G.D.M.)
*   Correspondence: luigi.tedone@uniba.it; Tel.: +39-0805442501

**Abstract:** Soil quality, nitrogen, and organic matter content are increasingly being researched due to their impact on the environment. We assessed the effects of different soil management practices on the distribution and accumulation of soil organic carbon (SOC) in a durum wheat–faba bean rotation system cultivated in a Mediterranean-type area of Southern Italy, over six years. The effects of three levels of soil disturbance—conventional tillage (CT), minimum tillage (RT), and no tillage—(NT) on the SOC and nitrogen (N) content at soil depths of 0–15, 15–30, 30–60, and 60–90 cm were compared in a long-term experiment starting in the 2009–2010 growing season. The three soil management systems showed significant differences ($p < 0.05$) in the surface layer (0–15 cm depth) in SOC content and total nitrogen, with the largest accumulation occurring in the conservation system (NT). In the deep layers (30–60 and 60–90 cm), however, no significant differences were found between the three tillage systems. The ascending order of the tendency to accumulate SOC and N in the soil in the 0–15 cm layer was NT > CT > RT. In addition, the C/N ratio showed a more equilibrated rate in the NT system. The conservation tillage (NT) gave the best results in terms of the physical characteristics of the soil, showing a higher stability index compared to CT and RT. Conservation tillage is therefore recommended for wheat cultivation in the dry areas of Southern Italy, due to its benefits in terms of both crop yield improvements and environmental protection.

**Keywords:** sustainability; crop rotation; soil management; organic carbon; nitrogen; soil structure

## 1. Introduction

The increasing interest in the use of sustainable agricultural practices to increase soil organic carbon (SOC) is directly correlated with the $CO_2$ concentration in the atmosphere and global warming.

Soil represents the biggest terrestrial SOC reservoir (~1580 Gt C) and is involved in several aspects of food production, such as soil fertility and human health [1,2].

Today, the agricultural sector is increasingly oriented towards the introduction of sustainable farming practices that can preserve resources, such as water and fertilizer inputs, while improving SOC and reducing greenhouse gas (GHG) emissions [3].

The adoption of sustainable systems of cultivation, such as conservation agriculture (CA) management practices, can maintain food security and enhance environmental sustainability [4].

Soil management practices play a very important role in soil quality by changing the SOC and nutrient content, in particular, the nitrogen (N) concentration, with a subsequent effect on the GHG emissions from the soil. Agricultural soils can be considered a net sink or source for greenhouse gases (GHGs) [5] depending on many factors, including soil management practices, cropping systems, and weather conditions. If proper management is applied, agriculture soil can act as a carbon storage system, thus helping to reduce GHG emissions [6,7].

Several studies have been carried out to evaluate the variations in SOC and N in soil under different soil management practices [8–10]. The environmental impact of traditional agricultural ecosystems has convinced many farmers to adopt more sustainable practices in place of conventional soil management systems. Traditional/conventional ploughing (CT), when performed in good humid soil conditions, may have immediate benefits, including a reduction in soil compaction, burial of crop residues, and the control of weed infestation. However, CT increases the risk of soil erosion, nutrient loss, and depletion of organic matter (SOM), thus affecting soil quality [11] and crop productivity in the long run. SOC accumulation depends on the balance between the quality and quantity of SOM, which is influenced by a combination of climate variables, physical soil properties, and soil management.

Maintaining or incorporating crop residues is important for the maintenance of SOM, enhancing soil physical properties, and improving its quality, leading to increased amounts of SOC and the accumulation of N [12]. The accumulation of SOC and total N is highly influenced by soil depth, the type of cropping system, and the specific climatic conditions of the cultivation site [13,14]. Although several studies have indicated that RT increases SOC, the accumulation of SOC can take a long time, and in some cases, the accumulation effect may fail to occur even after 10 years [15]. In general, under a soil conservation management system, the highest SOC accumulation can be found in the upper soil layer (5–10 cm) [16,17]. Baker et al. [10] found that under a soil conservation management system, with sufficient crop residues left on the soil surface, the accumulation of SOC occurs only in the 0–30 cm soil layer.

The accumulation of SOC and N under RT compared to CT has shown conflicting trends [13]. West and Post [18] found that RT (reduced tillage), compared to the CT system, may fix 0.57 t C ha$^{-1}$ year$^{-1}$, although this increase of SOC may not be achieved even after 5–10 years.

Other studies [19] showed a negligible effect of RT on the SOC accumulation in the first 2–5 years. Blanco-Canqui and Lal [16] found no differences in SOC and N accumulation between conventional and conservation systems. Dicgwatlhe [4] suggested caution regarding organic matter accumulation in fresh climate conditions. In warm and arid environments, however, with limited amounts of plant residues, Campbell et al. [20] found that the SOC accumulation in the soil can reach a maximum level in a relatively short period of time (5–6 years).

To better understand the dynamics and the SOC accumulation mechanism, it is necessary to define the key indicators for the choice between a conventional and conservation system [21].

Different soil management systems can affect the biological, chemical, and physical properties of the soil. For example, variations in apparent density, porosity, and soil temperature influence the accumulation of SOC [22]. In addition, tillage systems and residual management greatly affect the C/N ratio, which is an important indicator of soil quality [9,23]. In agricultural soil, the C/N ratio is closely related to the type of organic material and climatic conditions, and, therefore, is directly related to the amount of biomass left in the soil in the form of crop residues. Al-Kaisi et al. [24] found that the right crop management, together with RT, could improve soil C and N sequestration; they showed that the increase in SOC and total N was due to the decrease in the SOM mineralization rate. Soil management in reduced tillage can positively influence the structure, which is a component of soil fertility. The aggregates of the soil are more stable in a reduced soil management system than under traditional ploughing [25,26].

Reduced tillage systems are thought to improve both soil structure and SOM, especially in arid and semi-arid climates [27–30]. However, the success of conservation tillage in increasing SOM, SOC, and total N is related to the local soil type and climate conditions, and their effect on the structural stability, which is variable [31]. In fact, aggregate stability and SOM are two important indicators of soil quality, and they are closely linked and interdependent [32]. They are often mentioned as key indicators in assessing the fertility of

a soil and can also be regarded as an indicator of environmental protection, as well as the environmental sustainability of intensive management systems [33].

The SOM content plays an important role in the overall stability of soil, increasing its fertility [34,35], especially in semi-arid areas characterized by poor, compacted, and non-structured soils with a low level of C [35]. Several studies [36–38] have found positive effects of RT on the aggregation of soil systems and SOM content in different types of soils and climates. However, the total SOM may require a longer or shorter period to be modified by processing techniques [39–43], since SOM is very heterogeneous and consists of a complex mixture of organic compounds, with a variable composition and content [44].

Other authors [45] have reported that conservation agriculture (CA) may offer a potential solution for long-term nutrient management with a lower carbon footprint for soil organic carbon (SOC) sequestration. In a meta-analysis on the effects of crop rotation and NT on GHG emissions, pulse (legume) inclusion in monoculture cropping significantly mitigated the global warming potential by reducing synthetic nitrogen requirements and increasing SOC sequestration [46–49].

The intensive agriculture adopted in the last few decades has had a negative impact on the soil. Today, the adoption of conservation techniques is an interesting alternative for sustainable agriculture due to their ability to reduce input costs, labor, and working time, as well as environmental benefits, including energy saving and SOC sequestration [50].

However, few studies have been conducted on the mechanism of SOC accumulation related to soil tillage management in the Mediterranean environment of Southern Italy. A broad understanding of tillage management and the dynamics of SOC and N in the long term is needed [51].

The present study evaluated the effects of different soil tillage techniques on the chemical and physical properties of soil in a typical Mediterranean area of Southern Italy. The main components of fertility (SOM and its distribution, total N, the structural stability index of the soil, and the accumulation of organic carbon) were assessed under different soil management techniques in order to determine the best system for crop productivity and environmental protection.

## 2. Materials and Methods

### 2.1. Site Description

The study was carried out at an experimental education center of the University of Bari, in a typical Mediterranean area (Southern Italy), located in Policoro; 40°10′20″ N, 16°39′04″ E, in the Basilicata region (MT). This site is 15 m above sea level and is characterized by a Mediterranean climate according to the De Martonne classification. It has an average annual rainfall of 560 mm, distributed mainly in the autumn and winter, and overall mild temperatures, with a maximum temperature of 40–42 °C in the summer. The soil has a loamy texture according to the USDA classification system (Table 1).

**Table 1.** Physical and chemical properties of soil from the study site.

| Characteristics | M. Unit | Value |
|---|---|---|
| Total Nitrogen (Kjeldahl method) | $g\ kg^{-1}$ | 1.7 |
| Available P (Olsen Method) | $mg\ kg^{-1}$ | 27.6 |
| Nitrate (Nitrate Test Kit $NO_3^-$) | ppm | 14.3 |
| Organic matter (Walkley–Black method) | % | 2.8 |
| Total lime | % | 8.8 |
| Exchangeable Na (ESP) | % | 1.9 |
| pH | - | 7.72 |
| Exchangeable $K_2O$ (ammonium acetate method) | $mg\ kg^{-1}$ | 227 |
| Total carbonate | $g\ kg^{-1}$ | 15.0 |
| Active carbonate (Druineau) | $g\ kg^{-1}$ | 5.0 |
| Soil salinity (ECe) (saturated paste extract method) | $dS\ m^{-1}$ | 0.95 |

### 2.2. Experimental Design and Crop Management

A long-term open field experiment established during the 2009–2010 growing season was carried out, starting with a biannual rotation using durum wheat (*Triticum turgidum L. subsp. Durum*) cv. IRIDE, and faba bean (*Vicia faba var. Minor Persons.*) cv. Prothabat. A strip-plot design over 9 ha was established with three replicates, with a single plot of 1 ha. The main treatments consisted of three levels of soil disturbance, as follows:

- Conventional tillage (CT): ploughing at an average depth of 35–40 cm and a subsequent complementary operation for the seed-bed preparation.
- Reduced tillage (RT): minimum disturbance of the soil, using a sub-soiler for a vertical cut of the soil until 30 cm depth. The main tillage operation was also followed by a complementary operation for seed-bed preparation.
- Conservation/no tillage (NT): crops were directly sown in the soil to obtain appropriate seed coverage using a sod-seeder.

The crop residues in the three soil management systems were compared, partially removed from the soil surface after harvesting, and subsequently incorporated into the soil in CT and RT, while they were left on the soil surface in NT.

The application of mineral N fertilizer was calculated based on the wheat removal dose of a previous yield of 5.0 t ha$^{-1}$ and considering the residual fertility of the faba bean, and it was administered at a dose of 60 units of nitrogen, applied as urea, during tillering. Weed control was carried out at the end of the tillering stage using selective chemical active compounds for durum wheat cultivation: Mesosulfuron-Methyl, Iodosulfuron-Methyl-Sodium, and Mefenpir-Diethyl were applied in the wheat fields for narrow leaf weeds, and 2.4 D was applied for broadleaf weeds. In NT, in addition, one application of Glyphosate was carried out before sowing. In the faba beans, Bentazone for broadleaf weeds and Propaquizafop for narrow-leaf weeds were used.

Sowing was carried out using 400 seeds m$^{-2}$ for durum wheat, and 50 seeds m$^{-2}$ for faba beans. The sowing was performed during late November and early December each year utilizing a sod-seeder mod. IGEA 2700 SEMINASODO was used for NT, and a conventional seed drill (LAROCCA 14 FALC) for the CT and RT systems. Fertilization, performed in coverage, was carried out with urea. The harvest was carried out in June–July with a Wintersteiger Classic plot combine, on a sample area of 15 m$^2$.

### 2.3. Sampling and Soil Analysis

Each year, from 2010 to 2018, the soil was sampled for the durum wheat and faba beans two months after harvesting. The soil was sampled at depths of 0–15, 15–30, and 30–60 cm. The soil was air dried and then sieved using a 2 mm sieve, then soil organic carbon (SOC) and total nitrogen (N) were determined. The Walkley and Black method was used for SOC analysis, while total nitrogen in the soil was measured by the Kjeldahl method [52].

On the same samples from 2013 to 2018, the water stability index (WSI), an indicator of soil quality, was determined using a wet-sieving method described by Pagliai et al. (1997) [25,26]. For this purpose, air-dried soil aggregates (1–2 mm) were placed on a 0.25 mm mesh sieve and moistened by capillary action from a layer of wet sand, then immersed in deionized water at room temperature and shaken in alternating vertical movements (30 agitations/min$^{-1}$), prior to passing them through a sieve. The WSI was then calculated as follows:

$$(B - C)/((A \times K) - C)) \times 100$$

where A is the mass of air-dried soil aggregates, B is the oven-dry mass of aggregates remaining in the sieve, C is the mass of the sand fraction, and K is the correction factor for soil moisture content (k = mass of oven-dry aggregates divided by the mass of air-dry aggregates).

### 2.4. Statistical Analysis

All data were assessed by analysis of variance (ANOVA), considering the effect of year, soil management, and depth of sampling. We also considered the crop, as faba beans, like all legume species, have the ability to fix nitrogen, influencing the chemical and physical properties of the soil. In addition, mean data were compared using the Student Newman Keuls (SNK) at the $p \leq 0.01$ significance level using $p \leq 0.01$ significance level using CoStat 6.451 software (Monterey, CA, USA).

## 3. Results

For the statistical analysis, the ANOVA for type of cultivation was significant, as well as the nitrogen content, in consideration of the different abilities of cereals and legumes to fix nitrogen (Table 2).

**Table 2.** ANOVA measures related to the effect of cultivation on crop rotation.

| Effect | SOC | Total N | Water Stability Index (WSI) |
|---|---|---|---|
| Cultivations (wheat vs. faba beans) | * | *** | ** |

*, **, ***—non-significant or significant at $p \leq 0.05$, 0.01 and 0.001, respectively.

Regarding the results of cultivation, we considered the combination of year, tillage, and depth of sampling, and the relative combinations of the variables.

Considering the effect on faba beans, the three parameters considered were highly significant, as was the interaction. The effect on organic carbon was significant at $p \leq 0.01$ for tillage and soil depth sampling. Total N had high significance for soil depth sampling, while the water stability index was highly impacted by tillage (Table 3).

**Table 3.** ANOVA measures related to faba bean cultivation.

| | Soil Organic Carbon (SOC) | Total N | Water Stability Index (WSI) |
|---|---|---|---|
| Main effect | | | |
| Year (Y) | * | * | ** |
| Deep sampling (D) | ** | *** | ns |
| Tillage (T) | ** | * | *** |
| Interaction | | | |
| Y × D | * | ** | ns |
| Y × T | ** | *** | ** |
| D × T | ** | ns | ns |
| Y × D × T | ns | ns | ns |

*, **, ***—non-significant or significant at $p \leq 0.05$, 0.01, and 0.001, respectively.

Regarding the effect of the parameters on wheat cultivation seasons, soil organic carbon was affected by depth of sampling, $p \leq 0.001$, while tillage and year were significant at ≤0.05. The sampling depth was more significant than the other two parameters (total N and WSI) (Table 4).

### 3.1. Organic Matter Concentration and Stock of Organic Carbon (SOC)

The results revealed that the organic carbon concentrations varied significantly as a function of soil depth and soil management (Figure 1). The table shows that the effect of soil depth on the SOC data was clearer than the effect of soil tillage treatment across years. On average, the SOC values were significantly higher in the 30–60 cm soil profile, whereas the lowest values were reported in the first layer (0–15 cm) in the base year and in the 15–30 cm soil profile of the last growing season.

**Table 4.** ANOVA measures related to wheat cultivation.

| | Soil Organic Carbon (SOC) | Total N | Water Stability Index (WSI) |
|---|---|---|---|
| Main effect | | | |
| Year (Y) | ** | ** | * |
| Deep sampling (D) | *** | *** | *** |
| Tillage (T) | ** | * | * |
| Interaction | | | |
| Y × D | ** | * | ns |
| Y × T | ** | *** | ns |
| D × T | *** | ns | ns |
| Y × D × T | * | ns | ns |

*, **, ***—non-significant or significant at $p \leq 0.05$, 0.01, and 0.001, respectively.

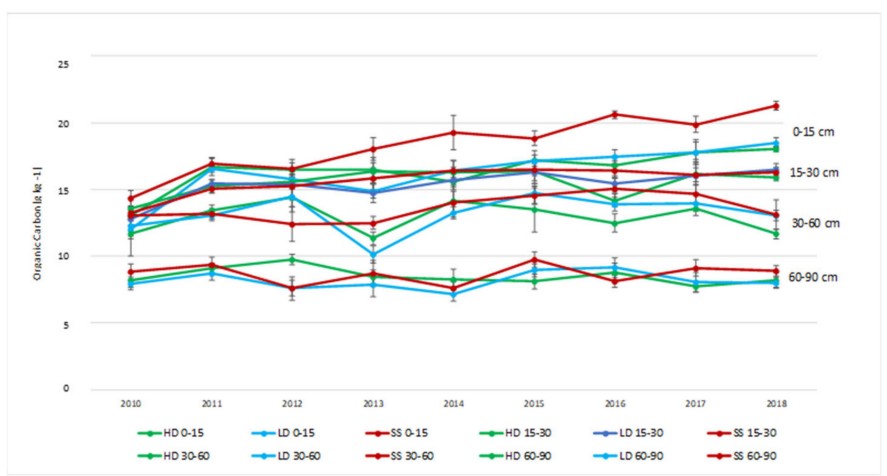

**Figure 1.** Evolution of soil organic carbon at different depths in the three soil management systems.

In 2010, in all tillage treatments, the SOC values of the differences between soil tillage were very low and were influenced by the sampling depth, with an average value of 13.0 g kg$^{-1}$ in both the 0–15 and 15–30 cm soil profiles, 13.0 g kg$^{-1}$ in the 30–60 cm soil profile, and 8.3 g kg$^{-1}$ in the deeper soil layer (60–90 cm).

The evolution, during the years of organic carbon content measurement, was very wide, with an increase from 11.9 to 14.0 g kg$^{-1}$ that was consistent in the 0–15 cm and 15–30 cm profiles, from an average of 13.1 g kg$^{-1}$ in 2010 to about 17.1 g kg$^{-1}$ in 2018. The other two profiles, 30–60 and 60–90 cm, showed no differences across the years, with mean values of 13.3 and 8.5 g kg$^{-1}$.

SOC values were recorded, with comparable values ranging between 12.0 and 13.6 g kg$^{-1}$ of soil, except for the first layer (0–15 cm), in the NT treatment. In the deeper layers (30–60 and 60–90 cm), the SOC values were also comparable across soil management treatments.

Comparing the results of the first year (2010) to those of the last year (2018), large differences were found that were higher in the 0–15 cm soil profile (17.05 g kg$^{-1}$) than in the 15–30 cm profile (15.53 g kg$^{-1}$). On the other hand, in the last two soil profiles (30–60 and 60–90 cm), the values were relatively stable between the first and the last years, with average SOC values of 13.06 and 8.46 g kg$^{-1}$ reported in the two layers, respectively.

The tillage system had a significant influence on SOC, with higher values in the NT system (Table 5).

**Table 5.** Organic carbon concentration (g kg$^{-1}$) evolution in conventional tillage (CT), reduced tillage (RT), and no-tillage (NT), at the 0–15, 15–30, 30–60, and 60–90 cm soil depths during 2010–2018. Different letters within each group indicate significant differences according to the Student Newman–Keuls (SNK) multiple-range test ($p \geq 0.05$).

| Variable | Organic Carbon | | |
|---|---|---|---|
| Year | g kg$^{-1}$ | | |
| 2010 | 11.76 [d] | ± | 0.23 |
| 2011 | 13.56 [b] | ± | 0.33 |
| 2012 | 13.45 [b] | ± | 0.36 |
| 2013 | 12.94 [c] | ± | 0.38 |
| 2014 | 13.69 [b] | ± | 0.42 |
| 2015 | 14.33 [a] | ± | 0.37 |
| 2016 | 13.61 [b] | ± | 0.43 |
| 2017 | 14.24 [a] | ± | 0.41 |
| 2018 | 14.13 [a] | ± | 0.45 |
| Tillage system | | | |
| CT | 13.35 [b] | ± | 0.36 |
| RT | 13.35 [c] | ± | 0.38 |
| NT | 14.10 [a] | ± | 0.42 |
| Soil depth | | | |
| 0–15 | 17.05 [a] | ± | 0.24 |
| 15–30 | 15.53 [b] | ± | 0.15 |
| 30–60 | 13.06 [c] | ± | 0.20 |
| 60–90 | 8.46 [d] | ± | 0.12 |

However, in the soil profile of 0–30 cm, an annual accumulation rate of 1.98 t ha$^{-1}$ year$^{-1}$ was found, which was higher in the two conservative systems (RT and NT) than the CT soil management systems (2.21 and 2.15 tha$^{-1}$year$^{-1}$ in RT and NT, respectively, in comparison with CT management, 1.58 t ha$^{-1}$year$^{-1}$) (Table 6).

*3.2. Concentration of Nitrogen*

The results for the N content in the soil showed that, despite the significant differences reported between the soil management treatments, the effect of soil management on the nitrogen stock in the soil was less pronounced compared to SOC. The table shows that the effect of soil depth, however, was more pronounced compared to the effect of the soil management treatments.

The results showed that the N concentration was higher in the upper layers, with average values ranging between 1.81 and 1.64 g kg$^{-1}$ in the 0–15 and 15–30 cm profiles, followed by the soil profile of 30–60 cm with a mean of 1.49 g kg$^{-1}$. In the last sampled soil profile (60–90 cm), the average value was 1.15 (Table 7).

In general, in the season of sampling after faba bean cultivation, the N values were higher at 1.61 g kg$^{-1}$ compared to 1.49 g kg$^{-1}$ after wheat cultivation, due to the fixation activity of faba beans leading to N enrichment due to nitrogen fixation.

The effect of year, related to the different behavior of the plants based on the seasonal conditions, was more variable in the faba bean season, with values that ranged between 1.45 and 1.82 g kg$^{-1}$, with respect to the wheat season, where the data ranged between 1.29 and 1.56 g kg$^{-1}$.

Tillage effect was also significant, with values between 1.54 and 1.66 g kg$^{-1}$ and higher values in CT and NT management, of 1.66 and 1.62 g kg$^{-1}$.

The effect of depth was wide, with data that ranged (faba beans and wheat) between 1.81 (0–15 cm depth), 1.68 (15–30 cm depth), 1.49 (15–30 cm depth), and 1.15 (30–60 cm depth).

The effect of tillage x depth was very important, with a higher concentration of nitrogen in the 0–15 and 15–30 cm in the conservative systems, in particular NT, with respect to the CT system.

**Table 6.** Mean soil organic carbon stock (2010–2020), net increase, and annual accumulation rate in a faba bean-wheat rotation under different tillage systems: conventional tillage (CT), reduced tillage (RT), and no-tillage (NT).

| Soil Depth (cm) | 2010 | | | 2018 | | | Net Increase 2010–2018 | | |
|---|---|---|---|---|---|---|---|---|---|
| | CT | RT | NT | CT | RT | NT | CT | RT | NT |
| | (t ha$^{-1}$) | | | (t ha$^{-1}$) | | | (t ha$^{-1}$) | | |
| 0–15 | 25.5 | 23.5 | 28.0 | 35.2 | 36.1 | 41.4 | 9.7 | 12.6 | 13.4 |
| 15–30 | 26.5 | 24.9 | 25.9 | 31.1 | 32.2 | 31.8 | 4.5 | 7.3 | 6.0 |
| 30–60 | 45.5 | 47.9 | 50.8 | 45.6 | 49.9 | 51.2 | 0.1 | 2.0 | 0.3 |
| 60–90 | 32.0 | 31.1 | 34.5 | 32.1 | 31.3 | 34.7 | 0.1 | 0.2 | 0.3 |
| Mean | 129.5 | 127.4 | 139.2 | 143.9 | 149.5 | 159.2 | 14.4 | 5.5 | 5.0 |
| Annual accumulation rate | | | | | | | | | |
| 0–30 | | | | | | | 1.58 | 2.21 | 2.15 |
| 30–90 | | | | | | | 0.02 | 0.25 | 0.07 |
| ANOVA | | | | | | | | | |
| Year | | | | | | | | *** | |
| Tillage | | | | | | | | *** | |
| Soil Depth | | | | | | | | *** | |
| Year × Tillage | | | | | | | | ** | |
| Year × Depth | | | | | | | | *** | |
| Tillage × Depth | | | | | | | | ** | |
| Tillage × Year × Depth | | | | | | | | ns | |

**, ***—non-significant or significant at $p \leq 0.05$, 0.01, and 0.001, respectively.

**Table 7.** Concentration total N (g kg$^{-1}$) during the period 2010–2018 in conventional tillage (CT), reduced tillage (RT), and no-tillage (NT) for the 0–15, 15–30, 30–60, and 60–90 cm soil depths. Different letters in each group indicated significant differences ($p \geq 0.05$).

| Cultivation | Variable | Nitrogen Content | | |
|---|---|---|---|---|
| | Year | g kg$^{-1}$ | | |
| wheat | 2010 | 1.51 [c] | ± | 0.040 |
| faba bean | 2011 | 1.67 [a] | ± | 0.044 |
| wheat | 2012 | 1.52 [c] | ± | 0.041 |
| faba bean | 2013 | 1.52 [c] | ± | 0.047 |
| wheat | 2014 | 1.44 [d] | ± | 0.043 |
| faba bean | 2015 | 1.52 [c] | ± | 0.044 |
| wheat | 2016 | 1.48 [cd] | ± | 0.039 |
| faba bean | 2017 | 1.59 [b] | ± | 0.040 |
| wheat | 2018 | 1.52 [c] | ± | 0.040 |
| | Tillage system | | | |
| | CT | 1.59 [a] | ± | 0.043 |
| | RT | 1.48 [c] | ± | 0.039 |
| | NT | 1.52 [b] | ± | 0.044 |
| | Soil depth | | | |
| | 0–15 | 1.81 [a] | ± | 0.028 |
| | 15–30 | 1.68 [b] | ± | 0.032 |
| | 30–60 | 1.49 [c] | ± | 0.035 |
| | 60–90 | 1.15 [d] | ± | 0.037 |

### 3.3. C/N Ratio

The C/N ratio is directly influenced by the amount of crop residue, the intensity of fertilization, and the rate of organic matter decomposition. Mechanical tillage can sig-

nificantly change this rate. The differences in C/N value can, therefore, be attributed to the different methods of soil management. Ploughing the soil, which mixes and incorporates crop residues into the soil, facilitates the decomposition of organic matter and its mineralization. In all cases, the value of C/N was under the indicative value of 10 for a typical Mediterranean country, where the temperature increases the mineralization activity. In the NT soil management systems, the C/N ratio was more equilibrated, which can be attributed to the higher percentage of crop residues in the upper layers that have not decomposed, as indicated by other studies, and to the more stable soil temperature, which reduces the mineralization of organic matter [23].

Additionally, values of C/N in the deeper layers (Table 4) appear to be attributed to the low concentration of both organic matter and nitrogen, but in particular N, due to the radical behavior of the crop [17]. In conservative systems, the roots tend to explore the outer layers of the soil, where most of the organic matter and crop residues are concentrated, which tends to maintain a constant humidity due to their mulching effect.

The C/N ratio increased over the years, from 8.02 in 2010 to 9.29 in 2018 (Table 4). With regards to the depth, the C/N ratio in the superficial layers (0–15 and 15–30 cm) was higher than in the 30–60 and 60–90 cm layers. The performance of the C/N ratio (the difference between the base and the last year of the experiment, 2010–2018) was always higher in the NT than the other two systems (Table 8).

**Table 8.** The C/N ratio during the period 2010–2018 in conventional tillage (CT), reduced tillage (RT), and no-tillage (NT) for the 0–15, 15–30, 30–60, and 60–90 cm soil depths. Different letters in each group indicated significant differences ($p \geq 0.05$).

| Cultivation | Variable | CN Rate | | |
|---|---|---|---|---|
| | Year | | | |
| wheat | 2010 | 8.02 [e] | ± | 0.134 |
| faba bean | 2011 | 8.41 [d] | ± | 0.200 |
| wheat | 2012 | 8.96 [cd] | ± | 0.172 |
| faba bean | 2013 | 8.69 [cd] | ± | 0.179 |
| wheat | 2014 | 9.67 [a] | ± | 0.254 |
| faba bean | 2015 | 9.61 [a] | ± | 0.154 |
| wheat | 2016 | 9.3 2 [ab] | ± | 0.163 |
| faba bean | 2017 | 9.01 [ab] | ± | 0.139 |
| wheat | 2018 | 9.29 [ab] | ± | 0.109 |
| | Tillage system | | | |
| | CT | 8.71 [c] | ± | 0.188 |
| | RT | 9.03 [b] | ± | 0.177 |
| | NT | 9.26 [a] | ± | 0.170 |
| | Soil depth | | | |
| | 0–15 | 9.01 [ab] | ± | 0.147 |
| | 15–30 | 8.98 [ab] | ± | 0.141 |
| | 30–60 | 9.16 [a] | ± | 0.217 |
| | 60–90 | 8.85 [b] | ± | 0.202 |

*3.4. Index of the Structure of the Soil (WSI)*

The analysis of the Water Stability Index (WSI) under different treatments confirmed the influence of soil management on this parameter, in combination with cultivation and seasonal course.

The effect of cultivation, already explained before, is another aspect able to influence structure stability [53] and was confirmed by the average values of the structural index after faba bean cultivation (14.4) with respect to wheat (13.2).

Considering the year effect, in wheat, higher values were registered in 2018, with an average value of 14.7, while in faba beans, 2017 was the year with higher values (15.8).

The effect of soil management was, in both cultivations, in favor of conservative soil management (NT), where the structural stability index was found to be 14.8% in

wheat and 15.9% in faba beans, while slightly lower results were found in the other two soil management types of 12.3–13.7 and 12.6–13.7 for CT and RT, and for wheat–faba beans, respectively.

Analyzing the three depths separately, we found that, in general, higher values were measured in the first layer, with 16.7 in wheat and 17.8 in faba beans, while the value of the structure decreased in the deeper layers (Table 9).

**Table 9.** Index of structure during the period 2013–2018 in conventional tillage (CT), reduced tillage (RT), and no-tillage (NT) for the 0–15, 15–30, 30–60, and 60–90 cm soil depths. Different letters in each group indicated significant differences ($p \geq 0.05$).

| Cultivation | Variable | CN Rate | | |
|---|---|---|---|---|
| | Year | | | |
| faba bean | 2013 | 13.7 [a] | ± | 0.475 |
| wheat | 2014 | 13.9 [a] | ± | 0.197 |
| faba bean | 2015 | 14.6 [a] | ± | 0.309 |
| wheat | 2016 | 14.2 [a] | ± | 0.307 |
| faba bean | 2017 | 13.9 [a] | ± | 0.310 |
| wheat | 2018 | 14.7 [a] | ± | 0.309 |
| | Tillage system | | | |
| | HD | 13.0 [b] | ± | 0.344 |
| | LD | 13.2 [b] | ± | 0.301 |
| | SS | 15.4 [a] | ± | 0.363 |
| | Soil depth | | | |
| | 0–15 | 15.1 [a] | ± | 0.388 |
| | 15–30 | 13.7 [b] | ± | 0.296 |
| | 30–60 | 12.7 [c] | ± | 0.328 |

## 4. Discussion

This study showed that the soil management system is crucial in influencing the chemical and physical properties of the soil.

In particular, the soil organic carbon and nitrogen content represent fundamental aspects for evaluating soil quality [54,55]. These components are influenced by tillage, with a modification of the rate of microbial decomposition of crop residues and N availability and a reduction in SOC content [56]. The SOC and N concentrations decreased with an increase in soil depth, depending on the type of tillage soil management. This has been confirmed by other studies [14,17] demonstrating the effect of organication of cultural residue on accumulation of SOC in the first layers, generally 0–35 cm, of the soil. The nitrogen content showed a similar behavior, with an equilibrium between these two components in the soil.

The effect of rotation was positive in all cases studied, as the data suggested an increase of SOC content from an average of 11.7 g kg$^{-1}$ in 2010 to 14.1 g kg$^{-1}$ in 2018 [57]. The data are interesting considering the environmental aspect of CO2 reduction in the atmosphere, with an accumulation of 8.3 Mg ha$^{-1}$ in the soil, as reported in Table 6.

On average, the best values were obtained under the NT treatment, with a carbon stock value of 14.4 Mg ha$^{-1}$, compared to 5.5 for RT and 5.0 for CT, while the lowest N concentration values were measured under the RT treatment, regardless of the soil depth. In the first soil layer (0–30 cm), conservation tillage (NT) also performed better than the other two soil management systems (RT and CT), as confirmed by other studies [52].

The N and SOC concentrations in the upper layers were attributed to the different soil impact of different tillage management, which led to soil mixing in the CT management, while there was a stratification in conservative management (NT).

The NT management was more effective than the other soil managements due to its ability to increase the SOC stocks in the 0–30 cm soil profile, which presents an important parameter for improving the soil structure. A higher SOC accumulation in the same soil profile can improve the soil structure, and therefore improve the fertility parameters of the soil [50,58,59].

The improvement in the water holding capacity in NT management is well known from different experiments. The cultural residue had a high content of SOM, leading to a better stability index, thus stabilizing aggregates in the upper layer of the soil, and protecting the deeper layers.

The accumulation of organic matter in the surface layers, as expected, led to the opposite behavior in the deeper layers, where the values were lower. In contrast, in the CT management, where the layers were mixed, the results clearly indicated a greater SOC accumulation in the deeper layers.

The increase in SOC over the years can also be attributed to the effect of the rotation, where the faba bean crop highlighted its positive effects both on the productivity of wheat in succession, and on improving the chemical and physical soil properties. Previous studies found that the intake effect of nitrogen fertilizer was irrelevant.

## 5. Conclusions

These results showed the effects of different soil management on the SOC, total N, and C/N rate stratification and distribution in the soil profiles. The SOC concentrations increased over the years in the conservation management (NT) of the soil, especially in the upper layers, compared to the conventional management, where the soil profile was mixed each year. Consequently, distinct layers of SOC were observed in the RT and CT soil management, while the C/N ratio was higher in the NT soil management. It should be noted that there was a compensation between the beneficial effects of conservative agriculture and the positive effects of ploughing. The structural stability index confirmed the effectiveness of conservative soil management (NT) compared to intensive management (CT), highlighting that the stability of the structure in conservative soil management cannot be achieved easily. This is because it seems to be very influenced by the soil type, the number of steps involved in the mechanical processes, and the weather conditions. The benefits associated with the conservation technique could be a long-term strategy to improve crop yield, quality of the soil, and environmental impacts, and thus achieve more sustainable conditions in agricultural systems.

**Author Contributions:** Conceptualization, G.D.M. and L.T.; methodology, L.T.; software, L.T.; validation, L.V., L.T. and G.D.M.; investigation, L.V.; data curation, L.V. and L.T.; writing—original draft preparation, L.T. and L.V.; writing—review and editing, L.T.; funding acquisition, G.D.M. All authors have read and agreed to the published version of the manuscript.

**Funding:** The research work was supported and funded by University of Bari A. Moro.

**Conflicts of Interest:** The authors declare no conflict of interest.

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
