# Peer review of "Effects of Different Types of Soil Management on Organic Carbon and Nitrogen Contents and the Stability Index of a Durum Wheat–Faba Bean Rotation under a Mediterranean Climate"

_agronomy, doi:10.3390/agronomy13051298_

Round 1

Reviewer 1 Report

Dear Editor,

Manuscript ID agronomy-2279064, titled: “Effects of different soil tillage on organic carbon, nitrogen con- 2 tents and stability index of a durum wheat-faba bean rotation 3 under Mediterranean climateauthors: Luigi Tedone *, Leonardo Verdini and Giuseppe De Mastro, is beautifully written. The general remarks in the paper are that the paper was much clearer:

Point 1: In Abstract an line 15 add full year and in line 22 delete therefore… Line 15 long-term experiment, started in the 2009-2010 growing season.

….line 22. Conservation tillage is therefore recommended for wheat cultivation in the dry areas of southern Italy due to its benefit in terms of 23 both crop yield improvements and environmental protection.

Point 2: In the introduction, please delete However in line 58 and Furthermore in line 91.

Line 58. However, The accumulation of SOC and 58 total N are highly influenced by soil depth, type of cropping system and specific climatic 59 conditions of the cultivation site (13, 14).

Line 91 Furthermore, The aggregates of the soil are more stable in a reduced soil man- 91 agement system than under traditional ploughing (25,26)…

Line 109. Different Other authors (45) reported that conservation agriculture (CA) ….

Point 3. Line 140 please add italic name of Triticum turgidum L. and Vicia faba….

Line 140 Experimental design refers a long-term open field experiment established during the 2009-2010 growing season, starting a biannual rotation was considered using durum wheat (Triticum turgidum L. subsp. Durum) cv. IRIDE-and faba bean (Vicia faba var. Minor Persons.) cv. Prothabat.

Point 4: In Results delete Infact ….in line 197

Line 197 Infact, The data of Anova between type of cultivation was significant, as table re- 197 ported, and referring the nitrogen content, in consideration of the different ability of cere- 198 als and legumes to fix nitrogen (Tab. 2).

Point 5 In Line 247 Table 5. Organic carbon concentration (g kg−1) for conventional tillage .. ..

changed , in point (.).

Line 255 In Table 6. Mean soil organic carbon stock (2010-2020),..... changed , in point (.).

Line 267 In Table 7. Concentration total N (‰): 2010-2020..... changed , in point (.).

Point 6: In line 298 in 3.3 C/N ratio in text delete In fact…..

Line 298 In fact, In conservative systems, the roots tend to explore the

Line 301 In Table 8. C/N ratio in the different soil management systems (2010-2018)...... changed , in

point (.).

Line 325 In Table 9. Add name table and changed , in point (.).

Point 7: Discusion In line 344 delete in fact …

In fact, The cultural residue present high content of SOM, led to a better stability index, thus stabilizing aggregates in the upper layer of the soil, and also protecting the deeper layers…

.

Point 9. Add next citate

Line 341. A higher SOC accumulation in the same soil profile can improve the soil structure and therefore improve the fertility parameters of the 342 soil (50-52).

51.   Ljubičić N., Popović V., Ćirić V., Kostić M., Ivošević B., Popović D., Pandžić M., El Musafah Seddiq, Janković S. Multivariate Interaction Analysis of Winter Wheat Grown in Environment of Limited Soil Conditions. Plants–Basel. 2021, 10, 3: 604; https://doi.org/10.3390/plants10030604,

52.  Kostić, M., Ljubičić, N., Ivošević, B., Radulović, M., Popovic S., Blagojević, D., Popović, V. Spot- based proximal sensing for field-scale assessment of winter wheat yield and economical production. Agriculture and Forestry, 2021, 67,1: 103-113. DOI: 10.17707/AgricultForest.67.1.09

Point 10: In Conclusions in Line 357 merge text

.

These results show the effects of different soil management on the SOC, total N and C/N rate stratification and distribution in the soil profiles. The SOC concentrations increased over the years in the conservation management (NT) of the soil, especially in the upper layers, compared to the conventional management, where the soil profile was mixed each year. Consequently, distinct layers of SOC were observed in the RT and CT soil management, while the C/N ratio was higher in the NT soil management. It should be noted that there is a compensation between the beneficial effects of conservative agriculture, compared to the positive effects that determine the ploughing. The structural stability index confirms the effectiveness of the conservative soil management (NT) compared to the intensive management (CT), highlighting that the stability of the structure in a conservative soil management cannot be achieved easily. This is because it seems to be very influenced by the soil type, the number of steps involved in the mechanical process, and the weather conditions. The benefits associated with the conservation technique can be seen in the long-term as a strategy to improve crop yield, quality of the soil, environmental impacts reduction and thus more sustainable condition in agricultural system.

I ask the authors to correct the paper in order to improve the quality.

Author Response

Thank you for the suggestions provided to improve the manuscript submitted to your attention. I'm sure the manuscript is definitely improved thanks to your suggestions. for English, the paper has undergone expert proofreading. Attached are details of the revisions made, in response to each observation

Reviewer 2 Report

Dear Authors,

Your manuscript deals the effect of tillage systems on SOC, total soil N, C/N and soil structural parameter (WSI) based on a long term field experiment.

The manuscript needs major revisions.

Many aspects should be considered and improved:

=> There is a mixture of nomenclatur (HD?, LD?, SS?, CT, NT, RT)

=> Line 50: Ploughing with mouldboard plough generate plough pans, which are highly compacted, if not on-land ploughing is used.

=> What was the N-fertilization level?

=> Line 195 and Table 2: replace cultivation with crop

=> The stock of C and N in the total soil depth (0-90 cm) should be should be presented as a sum (not average). This parameter should be dicussed with literature.

=> There are significant interactions of the main factors for some parameters. This interactions must be also explained, because the result of the significant main factor is irrelevant if there is a statistical interaction!!!

=> The discussion is superficial. The main results of the concentration and stock parameters for C and N must discussed with literature.

These are the main points for the improvement of your manuscript. 

Author Response

(The authors gave the same response as above.)

Reviewer 3 Report

General comments

The article describes the effects using contrasting soil tillage methods for 8 years in a wheat-faba bean rotation on important soil parameters (C and N content, and stability index) in a semi-arid environment in Southern Italy.  

The experimental design and assessments are scientifically sound and delivers important new information.

Although the main result obtained higher C and N content and stability index where n-tillage is used has been reported before; there are relatively few studies in which the effects of contrasting tillage systems have been monitored over longer periods (8 years), and there is very limited information especially from regions with Mediterranean/semi-arid climates.

The authors have also assessed the impact of growing a wheat/faba bean rotation on the same soil parameters, and this should be described more clearly, since it is also an important result and, in my opinion as important.

However, although the science is sound, the introduction, presentation of results and discussion needs major improvements.

Introduction

·         The text is too long and wordy

·         Consider moving some of the information into the discussions and use it to explain to what extent the results obtained in this study are consistent with or contradict previous studies/findings in similar or different pedoclimatic environments

Methods

Please describe (i) the crops grown/rotation and (ii) tillage system used in the experimental fields prior to the start of the experiment in detail. This information is essential to understand the changes in soil parameters resulting from using a wheat-bean rotation for 8 years.

Results

·         The results section is too wordy, in places impossible to understand and very unfocused

·         Data shown in the Tables and Figures do not need to be repeated in the text

·         If there are subsections this should start at the beginning of the results section

·         Tables 2, 3 and 4 only present p-values for main effects and interactions but no main effect means, this is meaningless because you cannot, for example, see which tillage treatment resulted in overall higher or lower values

·         Section 3.1/Figure 1; I assume that the SOC data described are means from all three tillage treatments; this needs to be stated very clearly or if my assumption is wrong it needs to be explained what the means/SE in Figure 1 show

o   If Figure 1 is based on SOC data means from all 3 tillage treatments, it shows the effect of growing a wheat-faba bean rotation for 8 years on SOC

o   This should be more clearly described

Recommendations

·         I strongly recommend that in the Tables showing results of the 3-factor ANOVAs with (i) year (2010 versus 2018), (ii) tillage system (CT, RT, NT) and (iii) depths (0-15, 30-60, 60-90) as factors, the authors report the main effect means ±SE (instead of mean for individual treatment combinations) together with p-values for the main effects and interactions.

·         I strongly recommend that results from 3-factor ANOVAs with (i) year (2010 versus 2018), (ii) tillage system (CT, RT, NT) and (iii) depths (0-15, 30-60, 60-90) as factors, are presented for all

parameters assessed (water stability index, soil C and N content, C/N ration, carbon stock and the soil structural stability index)

o   If assessments started at a later year than 2010 the first year in which assessment were made should be compared to the last year in which assessments were made after the same crop

·         For parameters where 3-factor ANOVA detects a significant 3-way interaction (e.g. for soil N and C?), I would recommend to carry out separate 2-factor ANOVAs for each soil depths with (i) year and (ii) tillage system as factors. This is likely to detect significant main effects or an interaction only for the top soil (0-15)

·         For parameters where 3-factor ANOVA does not detect a significant 3-way interaction I would investigate only the year x tillage interaction further 

·         It is not possible to accurately analyse effects of using the three different tillage systems in the two different crops, since they are grown in different years and seasonal differences in climatic conditions will confound the crop effects; I would therefore recommend to remove the results shown in Table 7.

·         Table 9 has no Title but I assume that assume soil structure index assessments only started in 2013 after faba beans. No ANOVA results are presented! If ANOVA were done and showed that there were no significant effects this should be stated, rather then describing non-significant differences

Specific corrections needed

Abstract

Line 20-21 Sentence is grammatically wrong/and therefore makes no sense. Please rephrase

Results

Table 5. the acronyms used for different tillage systems in the Table title and the Table itself are not consistent (see encircled acronyms in Table 5). Why do the authors use HD, LD and SS in the Table and CT, RT and NT in the title of Table 5

Results presented in this Table show the effect of tillage systems on organic carbon concentrations over time. They increase over time, but at a greater rate with no tillage compared with the standard and reduced tillage system.

Rather than comparing the mean carbon concentrations of different years it would make more sense to compared results obtained at the beginning (2010) and the end (2018) which would show the effect of 8 years of using a cereal/legume rotation (= difference between 2010 and 2018) and the effects of tillage.  

Table 6. explain acronyms CT, RT and NT in the Table legend

Table 7. the acronyms used for different tillage systems in the Table title and the Table itself are not consistent (see encircled acronyms in Table 7). Why do the authors use HD, LD and SS in the Table and CT, RT and NT in the title of Table 7.

Table 8. explain acronyms in the Table legend

Table 9. add title and explain acronyms used in the legend

Discussion

·         Needs major improvement and expansion.

·         Relate results more to the findings of previous studies

·         Should also focus on the beneficial effects of a wheat-bean rotation

·         The authors mention the positive effect of beans in the rotation on wheat performance; if they have published crop performance data these should be discussed in more detail

·         If the authors have grain yield data and these have not been published yet, I would recommend including them in this article

·         Previous studies have reported negative trade-offs between C-sequestration and yield or crop quality (e.g. increased Fusarium mycotoxin levels in wheat grain) from using no-tillage, such potential trade-offs should also be discussed in the context of the results reported here

Author Response

(The authors gave the same response as above.)

Round 2

Reviewer 3 Report

The article has been significantly improved, but there is a need to improve the format of some of the Tables in line with the journals requirements.